# FMRP Interacts with RARα in Synaptic Retinoic Acid Signaling and Homeostatic Synaptic Plasticity

**DOI:** 10.3390/ijms22126579

**Published:** 2021-06-19

**Authors:** Esther Park, Anthony G. Lau, Kristin L. Arendt, Lu Chen

**Affiliations:** Departments of Neurosurgery, Neuropsychiatry and Behavioral Sciences, Stanford University School of Medicine, 265 Campus Drive, Stanford, CA 94305-5453, USA; estherp@stanford.edu (E.P.); aglau13@gmail.com (A.G.L.); klarendt@stanford.edu (K.L.A.)

**Keywords:** retinoic acid signaling, homeostatic synaptic plasticity, FMRP, fragile X syndrome

## Abstract

The fragile X syndrome (FXS) is an X-chromosome-linked neurodevelopmental disorder with severe intellectual disability caused by inactivation of the fragile X mental retardation 1 (*FMR1*) gene and subsequent loss of the fragile X mental retardation protein (FMRP). Among the various types of abnormal synaptic function and synaptic plasticity phenotypes reported in FXS animal models, defective synaptic retinoic acid (RA) signaling and subsequent defective homeostatic plasticity have emerged as a major synaptic dysfunction. However, the mechanism underlying the defective synaptic RA signaling in the absence of FMRP is unknown. Here, we show that RARα, the RA receptor critically involved in synaptic RA signaling, directly interacts with FMRP. This interaction is enhanced in the presence of RA. Blocking the interaction between FMRP and RARα with a small peptide corresponding to the critical binding site in RARα abolishes RA-induced increases in excitatory synaptic transmission, recapitulating the phenotype seen in the *Fmr1* knockout mouse. Taken together, these data suggest that not only are functional FMRP and RARα necessary for RA-dependent homeostatic synaptic plasticity, but that the interaction between these two proteins is essential for proper transcription-independent RA signaling. Our results may provide further mechanistic understanding into FXS synaptic pathophysiology.

## 1. Introduction

The fragile X syndrome (FXS) is the most common genetic form of intellectual disability with impaired cognitive and social skills, anxiety and autistic-like behaviors [1,2]. FXS is caused by the loss of function mutations in the fragile X mental retardation 1 (*FMR1)* gene, which encodes the fragile X mental retardation protein (FMRP) [3]. The most common cause of FXS is the expansion of CGG repeats in the 5′ untranslated region of the *FMR1* gene resulting in hypermethylation and silencing of the gene, although other loss-of-function mutations in the FMR1 coding regions have also been identified in rare cases of FXS [3,4,5,6]. Clinical studies in FXS patients combining the high-resolution *FMR1* expression level analysis with neurobehavioral assessments indicate that FMRP expression in the brain is the ultimate factor determining the severity of the neurobehavioral phenotype in humans with FXS, including autism spectrum disorders (ASDs) [7]. Additionally, repeat-associated non-AUG-initiated (RAN) translation of expanded CGG repeats generates toxic proteins and contributes to neurodegeneration in fragile-X associated disorders [8]. FMRP is an RNA-binding protein that regulates the translation of specific mRNA targets. It also interacts with other RNA-binding proteins to regulate a variety of cellular functions such as translation, transport, and splicing of mRNAs [9,10,11,12,13,14]. FMRP is abundantly expressed in the brain and is involved in synaptic functions [15]. Indeed, many of the mRNA targets and interacting proteins of FMRP are known to regulate the synapse structure and function [10,12,16,17,18,19,20]. The altered synaptic function and plasticity have been reported in various regions of the *Fmr1* KO mouse brain, including altered basal excitatory and inhibitory synaptic transmission, phasic and/or tonic GABAergic inhibition, certain forms of LTP and metabotropic glutamate receptor (mGluR)-mediated LTD [21,22,23,24,25,26,27,28]. Recent reports further establish impaired homeostatic synaptic plasticity as another major synaptic plasticity phenotype in FXS [29,30,31].

Homeostatic synaptic plasticity is a mechanism by which neurons maintain network stability in response to prolonged changes in network activity deviating from its optimal dynamic range [32,33,34]. For example, the prolonged blockade of neuronal activity leads to a general increase in excitatory and a decrease in inhibitory synaptic strengths. At excitatory synapses, changes in synaptic strength are often mediated by an altered synaptic abundance of a major type of glutamate receptor, the α-amino-3-hydroxy-5-methyl-4-isoxazoleprprionic acid receptors (AMPARs). Two major molecular mechanisms have been described to drive changes in AMPAR synaptic expressions in homeostatic synaptic plasticity—a retinoic acid (RA)-mediated translation-dependent mechanism responsible for an early homeostatic adaptation [35], and a CaMKIV-mediated transcription-dependent mechanism for a late phase homeostatic compensation [36]. Interestingly, the loss of FMRP in both mouse and human neurons results in the loss of RA-dependent homeostatic synaptic plasticity, while having no effect on the late phase transcription-dependent homeostatic plasticity [29,30,31].

RA is a well-known developmental morphogen known for its role in neurogenesis and neuronal development [37]. Beyond its function in development, RA is also the key molecule mediating translation-dependent homeostatic synaptic plasticity in mature neurons [29,38,39]. Suppressed under normal synaptic activity, the synthesis of RA is triggered by a significant reduction in intracellular calcium levels (i.e., during prolonged blockade of excitatory synaptic transmission) [40,41]. RA receptors (RARα, RARβ, and RARγ) are nuclear receptors that regulate RA-dependent transcriptional events during development [37]. In the context of homeostatic plasticity, RARα, which is broadly expressed in adult brains, acts as a translational repressor that mediates the non-genomic actions of RA. Specifically, the distribution of RARα shifts from a strictly nuclear-localized pattern during development towards a more cytosolic-localized pattern in mature neurons [38,42]. Cytosolic RARα functions as a RNA-binding protein that suppresses the translation of specific mRNAs recognized by RARα in a sequence-specific manner [38]. This translational repression can be removed once newly synthesized RA binds to RARα, leading to the activation of dendritic synthesis of specific proteins, such as the AMPAR subunit GluA1 [38]. Increased expression of GluA1 in dendrites supports the homeostatic increase in synaptic strength in response to prolonged synaptic inactivity. Previous EM studies show that RA stimulation recruits RARα to FMRP-containing granules, and subsequent GluR1 synthesis is triggered in these granules [42]. In *Fmr1* KO neurons, the loss of FMRP abolishes the RA-induced increase in protein synthesis and subsequent homeostatic changes at both excitatory and inhibitory synapses [30], indicating that FMRP plays a major role in RA-dependent homeostatic synaptic plasticity. However, it remains unclear how FMRP and RARα interact functionally to regulate the protein synthesis process required for RA-induced homeostatic synaptic plasticity. 

In this study, we investigated the physical interaction between FMRP and RARα and found that RARα directly binds to FMRP, and that the binding is enhanced in the presence of RA. Moreover, we established that this direct interaction is critical for RA-induced homeostatic synaptic plasticity. 

## 2. Materials and Methods

### 2.1. Animals

All breeding colonies were maintained in the animal facility at Stanford Medical School following standard procedures approved by the Stanford University Administrative Panel on Laboratory Animal Care. *Fmr1* knock-out mice (JAX: 004624 RRID:IMSR_JAX:004624) in the FVB background were obtained from The Jackson Laboratory. The RARα floxed mice were a gift from Drs. Pierre Chambon and Norbert Ghyselinck (IGBMC, Strasbourg, France) [43]. 

The *Fmr1*^−/y^::RARα^fl/fl^ double mutant mice were generated by crossing the *Fmr1* knock-out mice with the RARα^fl/fl^ mice. Genotyping was performed with the following primer sets. For RARα WT, Primer 1 (fwd) 5′-GTGTGTGTGTGTATTCGCGTGC-3′, Primer 2 (rev) 5′-ACAAAGCAAGGCTTGTAGATGC-3′; for RARα flox, Primer 1 with Primer 3 (rev) 5′-TACACTAACTACCCTTGACC-3′; for FMRP WT, WT (fwd): 5′-GTGGTTAGCTAAAGTGAGGATGAT-3′, WT (rev): 5′-CAGGTTTGTTGGGATTAACAGATC-3′; and for FMRP KO, KO (fwd) 5′-CACGAGACTAGTGAGACGTG-3′ and KO (rev) 5′-CTTCTGGCACCTCCAGCTT-3′.

### 2.2. Primary Hippocampal Cultures

Primary hippocampal cultures were made from postnatal day 0 (P0) or P1 *Fmr1*^+/y^::RARα^fl/fl^ and *Fmr1*^−/y^::RARα^fl/fl^ mice of both sexes as previously described [30,44]. The primary cultured hippocampal neurons were maintained in Neurobasal media supplemented with B-27 and Glutamax (Invitrogen, Waltham, MA, USA). The neurons were transduced with lentivirus at DIV5 or 6 and harvested at DIV14–15. 

### 2.3. Constructs

Recombinant protein expression constructs: All RARα constructs were cloned from rat sequences into pGEX-KG. Construction of full-length GST-RARα, GST-DBD, and GST-LBD was previously described [38]. The truncations of GST-DBD were PCR amplified from full-length rat RARα and cloned into pGEX-KG using BamHI and EcoRI: For GST-C/D, nucleotides 241–567; for GST-tC/D, nucleotides 361–567; and for GST-D, nucleotides 469–567. The FMRP-6XHis construct was amplified by PCR from FMRP-GFP and cloned into pET28a (Novagen, Madison, WI, USA) using EcoRI and NotI. The blocking peptide (nucleotides 361–567 of GST-RARα) and control peptide (nucleotides 4–210 of GST-RARα) were amplified by PCR from full-length rat RARα and cloned in pET28a (Novagen, Madison, WI, USA) using BamHI and EcoRI. 

Lentiviral constructs: The FMRP-GFP, I304N-GFP, and ΔRGG-GFP in pJHUG were previously described [30]. The truncation mutants of FMRP in pJHUG were PCR amplified using FMRP-GFP as a template and the following nucleotides were cloned into pJHUG using EcoRI and NotI; for 128–614-GFP, nucleotides 382–1842; for 218–614-GFP, nucleotides 652–1842; for 438–614-GFP, nucleotides 1318–1842; and for 1–219-GFP, nucleotides 1–654. 

The lentiviral plasmids expressing the blocking peptide or control peptide fused to the C-terminus of GFP were constructed by inserting the same nucleotides from the His-tagged recombinant protein expression plasmids described above into pFUGW using BsrGI and EcoRI. 

Flag-RARα was cloned by EcoRI into the L309 already containing IRES-GFP-CRE (gift from Dr. Thomas Sudhof’s lab) for lentiviral expression of Flag-RARα and GFP-CRE under human synapsin promoter (SYN-Flag-RARα-IRES-GFP-CRE). 

Mammalian expression constructs: The RARα-GFP in pEGFP-N1 was previously described [39]. The RARαΔtC/D-GFP was constructed by overlap PCR from rat RARα to delete nucleotides 361–567. The resulting fragment was cloned into pEGFP-N1 (Clontech, Mountain View, CA, USA) using XhoI and EcoRI. 

### 2.4. Fusion Protein Production

GST purification: The GST-RARα and truncation fusion proteins were isolated as previously described [38]. Briefly, the GST constructs were transformed into BL21(DE3) cells (NEB, Ipswich, MA, USA). Cells were grown to OD 600 of 0.5 then induced with 1 mM IPTG (Goldbio, Saint Louis, MO, USA) for 2–4 h at 37 °C. The cells were pelleted and lysed in the lysis buffer (20 mM sodium phosphate 7.4, 150 mM NaCl, 1% TX-100, protease cocktail (Roche, Indianapolis, IN, USA) containing lysozyme and universal DNAse (Pierce, Waltham, MA, USA)) at 4 °C for 30 min. Debris was cleared by centrifuging at 15,000 rpm for 20 min. The supernatant was then incubated with glutathione sepharose beads (Amersham, Piscataway, NL, USA) end over end at 4 °C for 2 h. The beads were then washed 5 times with a lysis buffer. Protein expression and purification was confirmed by SDS-PAGE followed by Coomassie staining or Ponceau S staining.

His-tagged purification: Isolation of FMRP-His tagged recombinant protein was the same as above except the FMRP lysis buffer was composed of 10 mM HEPES pH 7.5, 300 mM LiCl, 5 mM β-mercaptoethanol, 10 mM imidazole, 5% glycerol and protease inhibitor cocktail pellets (Roche, Indianapolis, IN, USA), and the supernatant was incubated with Ni-NTA agarose (Qiagen, Redwood City, CA, USA) rather than glutathione-sepharose beads. The beads were washed 3 times with the FMRP-lysis buffer containing 100 mM imidazole (rather than 10 mM). Purified peptides were eluted from beads with 500 mM imidazole. Protein expression, purification, and elution were confirmed by SDS-PAGE followed by Coomassie staining or Ponceau S staining.

Isolation of the blocking and control His-tagged peptides was performed as above using the following His-lysis buffer (50 mM NaH_2_PO_4_ pH 8.0, 300 mM NaCl, 10 mM imidazole, and protease inhibitors (Roche, Indianapolis, IN, USA)) and incubated with Ni-NTA agarose (Qiagen, Redwood City, CA, USA) for purification. The beads were washed 3 times with the His-lysis buffer containing 100 mM imidazole (rather than 10 mM). Purified peptides were eluted from beads with 500 mM imidazole. Protein expression, purification, and elution were confirmed by SDS-PAGE followed by Coomassie staining.

### 2.5. GST Pull-Down

Equal amounts of GST-purified recombinant proteins on glutathione-sepharose beads were incubated with either HEK cell lysate, lysate from *Fmr1* KO mouse embryonic fibroblast (MEF) cells or with equal amounts of purified T7-FMRP-His at 4 °C for 1 h end over end. The beads were then washed with the lysis buffer 5 times before eluting the protein in the 1X sample buffer for Western blot analysis. 

### 2.6. Immunoprecipitation

Primary hippocampal neuron IP: Fmr1^+/y^::RARα^fl/fl^ or Fmr^−/y^::RARα^fl/fl^ mice were used to culture primary hippocampal neurons. At DIV 5–6, the neurons were infected with lentivirus expressing Flag-RARα and CRE-GFP to knockout endogenous RARα. The infected neurons were harvested on DIV 14-15. The neurons were first washed with 1X PBS and then lysed for 30 min at 4 °C in the IP lysis buffer (50 mM Tris pH 7.3, 150 mM NaCl, 1% Triton X-100, 1 mM NaF, 1 mM sodium orthovanadate (NEB, Ipswich, MA, USA), and protease inhibitor cocktail 2 (Sigma, Saint Louis, MO, USA)). After lysing, samples were centrifuged at 14,000 rpm for 15 min at 4 °C. Then, the supernatant was removed and incubated with dynabeads (Invitrogen, Waltham, MA, USA) to preclear the lysate for 1 h at 4 °C. The precleared lysate was incubated with a M2 Flag antibody (mouse, Sigma, Saint Louis, MO, USA) or FMRP antibody (rabbit, Abcam, Cambridge, MA, USA) for 1 h at 4 °C, to which dynabeads (Invitrogen, Waltham, MA, USA) were added and further incubated at 4 °C for 1 h. The beads were washed 5 times with the IP lysis buffer. The proteins were eluted from the beads with the 1X SDS loading buffer then run on SDS-PAGE for Western blot analysis. 

HEK cell IP: HEK293T cells were cultured in 6-well dishes and transfected with 1 μg of pEGFPN-1 or RARα-pEGFPN-1 by calcium phosphate. The media was changed after 4 h. The following day the transfected HEK293T cells were treated with 1 μM of RA (Sigma, Saint Louis, MO, USA) for 1 h at 37 °C before harvesting. The cells were washed 2 times with PBS before harvesting in the IP-lysis buffer. In addition, 1 μM RA was added to the IP-lysis buffer for RA treated samples throughout the experiment to maintain a constant level of RA. After centrifuging, the lysate was incubated for 1 h with anti-GFP magnetic beads (MBL International, Woburn, MA, USA) at 4 °C. The beads were washed 5 times with the IP lysis buffer and proteins were eluded with a 1X SDS-loading buffer before the Western blot analysis. 

### 2.7. Western Blot 

Samples were run on 7%, 10%, 4–15% or 12.5% SDS-PAGE and transferred to either nitrocellulose or PVDF membranes. The membranes were blocked with 5% milk in TBS-T (1X Tris buffered saline containing 0.3% Tween-20) for 30 min at room temperature. Membranes were then incubated with a primary antibody diluted in 5% milk for 1 h at room temperature or 4 °C overnight. The primary antibody was washed 3 times for 10 min each with TBS-T, then incubated in the secondary diluted in 5% milk for 1 h at room temperature. The secondary was washed 3 times for 10 min each with TBS-T. The signal was detected by either ECL followed by development with film or scanning on Licor for the quantitative analysis. 

### 2.8. Antibodies

The following monoclonal antibodies were used in this study: FMRP 1C3 (WB, 1:1000, Millipore, Hayward, CA, USA), FMRP 7G1-1 (WB, 1:500, DSHB), and Flag (IP, Sigma, Saint Louis, MO, USA). The following rabbit polyclonal antibodies were used in this study: Flag (WB, 1:1000, Sigma, Saint Louis, MO, USA), GFP (WB, 1:5000, Abcam, Cambridge, MA, USA), and FMRP (WB, IP, 1:1000, Abcam, Cambridge, MA, USA). A chicken polyclonal antibody for GFP (WB, 1:5000, Abcam, Cambridge, MA, USA) was also used. The antibody for T7 was directly conjugated to HRP (WB, 1:5000, Millipore, Hayward, CA, USA). Anti-mouse IgG-HRP, anti-rabbit IgG-HRP, and protein-HRP (1:5000, Jackson Labs, West Grove, PA, USA) were used for ECL detected Western blots. For Licor imaging, IRDye anti-mouse IgG-800CW and IRDye anti-rabbit IgG-680RD was used (1:10,000, Licor, Lincoln, NE, USA). 

### 2.9. Organotypic Hippocampal Slice Cultures

Organotypic slice cultures were prepared from young RARα conditional knockout mice (P6 to P7) and placed on semi-porous membranes (Millipore, Hayward, CA, USA) for 10 to 12 days prior to recording [45,46]. Briefly, slices were maintained in a MEM based culture media comprised of 1 mM CaCl_2_, 2 mM MgSO_4_, 1 mM L-glutamine, 1 mg/L insulin, 0.0012% ascorbic acid, 30 mM HEPES, 13 mM D-glucose, and 5.2 mM NaHCO_3_. Culture media was a pH of 7.25 and the osmolarity was 320. Cultures were maintained in an incubator with 95% O_2_/5% CO_2_ at 34 °C. 

### 2.10. Viral Vectors and Viral Infection

Recombinant lentivirus was produced as previously described [39,47]. Briefly, using calcium phosphate, HEK293T cells were transfected with four plasmids, the lentiviral shuttle vectors (pVSVg, pMDL, and pREV) and transfer vector. The HEK293 culture media was collected 40–44 h after transfection and filtered with the 0.45 μm PVDF filter (Millipore, Hayward, CA, USA) to remove cellular debris followed by centrifugation through a 20% sucrose cushion at 50,000x g to concentrate the virus. The concentrated virus was dissolved in a small volume of PBS and stored at −80 °C. Lentiviruses expressing either the blocking peptide or control peptide fused to GFP [48] were injected into the CA1 region of organotypic hippocampal slice cultures at DIV 0 and expressed for 10 days. The viral injection was carried out using a picospritzer with 18-20 PSI coupled to a micropipette with a resistance of 8–10 MOhms. The virus was delivered into the CA1 cell body layer at 4–5 injection sites with 4–5 pulses (5 ms duration) per injection site. Recordings from the infected neurons and their uninfected neighboring neurons were carried out at DIV 10–12. 

### 2.11. Electrophysiology

Voltage-clamp whole-cell recordings were obtained from CA1 pyramidal neurons that were either uninfected or sparsely infected with constructs expressing control and blocking peptide fused to GFP, treated with either vehicle controls or 10 µM RA (for 4 h prior to recording), under visual guidance using transmitted light illumination. The vehicle control and RA treated cells were obtained from the same batches of slices on the same experimental day. A spontaneous miniature transmission was obtained in the presence of 1 μM of TTX and 100 μM picrotoxin in the external solution. For slices previously exposed to RA, the slices were washed out prior to recording. 

The recording chamber was perfused with 119 mM NaCl, 2.5 mM KCl, 4 mM CaCl_2_, 4 mM MgCl_2_, 26 mM NaHCO_3_, 1 mM NaH_2_PO_4_, 11 mM glucose, 0.1 mM picrotoxin, and 1 μM TTX at pH 7.4, gassed with 5% CO_2_/95% O_2_ and held at 30 °C. Patch recording pipettes (3–6 MOhms) were filled with 115 mM cesium methanesulfonate, 20 mM CsCl, 10 mM HEPES, 2.5 mM MgCl_2_, 4 mM Na_2_ATP, 0.4 mM Na_3_GTP, 10 mM sodium phosphocreatine, and 0.6 mM EGTA at pH 7.25. 

All electrophysiological recordings were carried out with Multiclamp 700B amplifiers (Axon Instruments, Perth, Western Australia) and the analysis was completed in Clampfit 10.2 (Molecular Devices, San Jose, CA, USA), MiniAnalysis version 6 (Synaptosoft, Decatur, GA, USA), Excel (Microsoft, Redmond, WA, USA), and Prism 10 (GraphPad, San Diego, CA, USA). 

### 2.12. Statistical Analysis 

All graphs represent average values ± SEM. Statistical differences were calculated according to parametric (for datasets with normal distribution) or nonparametric (for datasets that do not conform to normal distribution) tests using GraphPad Prism 9.0 software. Comparisons between multiple groups were performed with the Kruskal-Wallis ANOVA. When significant differences were observed, p-values for pairwise comparisons were calculated according to two-tailed Mann-Whitney tests (for unpaired data). For RA-mediated effects on interaction, p-values were calculated by the student’s *t*-test. All experiments were repeated at least 3 times.

## 3. Results

### 3.1. RARα Interacts Directly with FMRP 

We have shown previously that synaptic inactivity-triggered RA synthesis is normal in *Fmr1* KO mouse neurons [30], suggesting that the impaired RA-dependent homeostatic synaptic plasticity in these neurons is likely caused by alterations in the signaling pathway downstream of RA. Given that both RARα and FMRP are RNA-binding proteins that can be localized to the same RNA granules [42], we wondered whether they interact directly. First, in reduced systems, incubation of purified bacterial-expressed GST-RARα fusion protein with HEK cell lysates successfully pulled down endogenous FMRP from HEK cells (Figure 1A) and from P30 mouse hippocampal lysate (Figure 1B). Additionally, purified T7- and His-tagged FMRP was found associated with purified GST-RARα in an in vitro binding assay, suggesting that this interaction is indeed a direct binding between the two proteins (Figure 1C). 

Next, we tested whether the FMRP-RARα interaction occurs in hippocampal neurons. To circumvent the problem of lacking a reliable and specific antibody against RARα, we took advantage of the conditional knockout RARα mice (RARα^fl/fl^), in which molecular and functional replacement of endogenous RARα with an epitope-tagged RARα can be achieved [49]. Moreover, we used the constitutive *Fmr1* KO mice as a control for nonspecific binding that could occur in immunoprecipitation assays. We cultured primary hippocampal neurons from *Fmr1*^+/y^::RARα^fl/fl^ and *Fmr1*^−/y^::RARα^fl/fl^ mice. Transducing these cultured neurons with a lentivirus expressing both a Flag-tagged RARα and a GFP-tagged Cre recombinase driven by the human Synapsin promoter led to expression of Flag-RARα in the absence of endogenous RARα in hippocampal neurons with WT or FMRP KO background. Immunoprecipitation with anti-FMRP antibodies pulled down FMRP as well as RARα in the WT but not *Fmr1* KO neurons (Figure 1D), indicating high specificity of the FMRP antibody. Likewise, immunoprecipitation with anti-Flag antibodies pulled down RARα as well as FMRP in only *Fmr1* WT neurons (Figure 1E). Taken together, these data demonstrate that RARα interacts directly with FMRP in hippocampal neurons. 

### 3.2. Identification of Domains Necessary for RARα-FMRP Binding 

Among the six distinct domains within the RARα protein, the A/B and C domains are important for transcriptional activity and for DNA binding (Figure 2A) [37]. The D domain contains the nuclear localization signal (NLS) and part of the hinge region (Figure 2A). The E domain binds to retinoic acid and the F domain is critical for mRNA binding (Figure 2A) [38]. To determine which regions are important for RARα binding to FMRP, we first generated GST-fusion proteins of two RARα truncation mutants that include the DNA binding domain (DBD) or the ligand-binding domain (LBD) (Figure 2A). Incubation of these purified GST-RARα truncation mutants with P30 mouse hippocampal lysates demonstrated a clear binding between FMRP and RARα DBD but not the LBD domain (Figure 2B). Further dissection of subregions within DBD using additional RARα truncation mutants isolated the amino acids 120–155 of RARα as the critical region for its interaction with FMRP (Figure 2B). 

To further verify the critical requirement of the identified region in RARα for binding, we expressed GFP-tagged wildtype RARα or RARα-ΔtC/D (Δ120–189 a.a.) in HEK cells and examined their co-immunoprecipitation with endogenously expressed FMRP. As expected, both GFP-RARα and RARα-ΔtC/D were abundantly present in the protein complex pulled down by a GFP antibody, but FMRP was only present in complexes pulled down from GFP-WT-RARα-expressing cells, indicating that deletion of this region blocked the interaction between RARα and FMRP (Figure 2C).

Next, we investigated which domains of FMRP are essential for its interaction with RARα. FMRP consists of five domains that are critical for its function. Both KH-domains and the RGG box have been shown to have RNA-binding activity. The other two domains, the N-terminal domain of FMRP (NDF) and the protein-protein interaction domain (PPID), are known to mediate the FMRP interactions with other proteins such as CYFIP1, NuFIP2, FXR1, etc. [18,50,51,52] (Figure 3A). Thus, we generated various FMRP truncation mutants and point mutations (Figure 3A) and expressed them in mouse embryonic fibroblasts (MEFs) generated from *Fmr1* KO mouse (hence no interference from endogenously expressed WT FMRP). MEF lysates were incubated with purified GST-RARα in a pull-down assay. Previous work has shown that mutations that abolish the FMRP RNA binding activity and its association with polyribosomes (e.g., I304N mutation in KH2 RNA-binding domain or RGG box deletion) disrupt RA-mediated homeostatic synaptic plasticity [30,53,54,55,56,57]. Surprisingly, both FMRP-∆RGG and FMRP (I304N) were still able to bind to RARα. However, deletion of the PPID abolished the interaction of FMRP with RARα (Figure 3B). Moreover, a truncated form of FMRP containing only the NDF and PPID was able to interact with RARα (Figure 3B), suggesting that the PPID is necessary and sufficient for the FMRP interaction with RARα. 

### 3.3. RA Enhances the Interaction between RARα and FMRP 

Our previous EM study showed that RARα is recruited to FMRP-containing dendritic granules shortly after RA stimulation [42], raising the possibility that RA may influence the interaction between RARα and FMRP. We tested this in HEK cells expressing endogenous FMRP and a N-terminal GFP-tagged RARα. Immunoprecipitation with a GFP antibody pulled down RARα as well as FMRP in GFP-RARα expressing but not GFP expressing control HEK cells (Figure 4A). Treating the HEK cells with 1 μM RA for an hour significantly increased the pull-down efficiency (Figure 4B), indicating that RA enhances the interaction between FMRP and RARα. 

### 3.4. Interaction between FMRP and RARα Is Critical for RA-Dependent Synaptic Plasticity 

What might the FMRP-RARα interaction be important for? Previous studies showed that in *Fmr1* KO mouse neurons, the lack of FMRP expression leads to a complete loss of RA-dependent homeostatic synaptic plasticity [29,30]. In these studies, impaired homeostatic synaptic plasticity in FMRP-∆RGG- and I304N-expressing FMRP KO neurons indicate that signaling pathways downstream of FMRP/RARα binding were required for RA-dependent synaptic plasticity since these two FMRP mutants have normal binding to RARα. However, this does not exclude the possibility that normal FMRP/RARα binding is also required for RA-dependent synaptic plasticity. Here, we sought to test whether blocking the FMRP-RARα interaction affects RA-dependent synaptic plasticity. 

We generated a His-tagged peptide of the FMRP binding region in RARα to block the FMRP-RARα interaction (blocking peptide, Figure 5A). A His-tagged control peptide, generated with a sequence outside of the FMRP binding region and of similar molecular size, served as a control. The addition of purified bacterial-expressed blocking peptide to the mixture of purified GST-RARα and T7-FMRP-His significantly reduced the interaction between RARα and FMRP compared to the interaction in the presence of the control peptide (Figure 5B). 

After verifying the efficacy of the blocking peptide, we infected CA1 pyramidal neurons in the cultured wild type hippocampal slices with lentiviruses expressing GFP-tagged (to facilitate visualization of infected neurons) control and blocking peptides and examined the RA-induced increase in excitatory synaptic transmission in these neurons. Consistent with previous reports [39,45], the acute RA treatment (10 μM, 4 h) induced a rapid increase in excitatory synaptic responses in uninfected neurons as well as those infected with the control peptide, evidenced by a significant increase in the amplitude of miniature excitatory postsynaptic responses (mEPSCs) (Figure 5C,D). By contrast, neurons expressing the blocking peptide failed to respond to the RA treatment (Figure 5C,D). Thus, we established that the FMRP-RARα interaction is essential for RA-induced synaptic plasticity. 

## 4. Discussion

Synaptic RA signaling has been established as a major signaling pathway mediating synaptic inactivity-induced homeostatic synaptic plasticity [35,39]. Prolonged synaptic silencing triggers RA synthesis and RARα-mediated local translation of proteins essential for the expression of homeostatic synaptic plasticity. Previous studies suggest that FMRP may be critically involved in this form of homeostatic synaptic plasticity as in *Fmr1* KO mouse neurons and fragile-X syndrome patient-derived human neurons, where RA-dependent homeostatic synaptic plasticity is absent [29,30,31]. In this study, we investigated the FMRP-RARα interaction as a potential molecular requirement for RA-dependent homeostatic synaptic plasticity in neurons. Using both in vitro and in situ biochemical binding assays, we demonstrated that FMRP and RARα directly bind to each other. Systematic mapping of the binding domains in both proteins allowed for us to identify regions critical for the direct interaction of RARα and FMRP. Subsequently, the generation of a peptide that blocks this interaction impaired homeostatic synaptic plasticity. Thus, we established that the FMRP-RARα interaction is required for RA-induced synaptic plasticity. 

During RA-dependent homeostatic synaptic plasticity, a reduction in dendritic calcium levels due to reduced synaptic activity results in a decrease in calcineurin activity, which triggers RA synthesis [40,41]. Downstream of RA synthesis, complex molecular and cellular processes occur to realize multiple aspects of homeostatic adjustment at both pre- and postsynaptic sites. These processes include, but are not limited to, RA-induced disinhibition of protein synthesis from RARα-bound mRNAs, exocytosis of AMPA receptor (AMPAR)-containing vesicles and synaptic delivery of AMPARs, endocytosis of GABA_A_ receptors (GABA_A_R), and enhancement of presynaptic release probability through retrograde signaling [29,38,39,41,45]. The discovery of the requirement of FMRP in RA-dependent homeostatic synaptic plasticity adds additional complexity to the signaling pathways [29,30,31]. The RA synthesis in response to the activity blockade has been shown to be intact in FMRP KO neurons [30], placing the FMRP involvement at a step downstream of RA synthesis. Given that the major role of FMRP is an mRNA-binding protein and a translational regulator [9,10,11,17], and synaptic RA signaling requires RARα-mediated local protein synthesis, it is perhaps not surprising that FMRP mutants that are impaired in translational regulation of target mRNAs (FMRP-∆RGG and I304N) fail to rescue RA-dependent homeostatic synaptic plasticity [30]. However, results from this study indicate that the FMRP involvement in synaptic plasticity may be multi-layered, and that a direct binding between FMRP and RARα is necessary (albeit not sufficient since both ∆RGG- and I304N mutants of FMRP have normal binding to RARα). 

What might the FMRP-RARα interaction be required for? One possibility is protein trafficking and dendritic targeting. Similar to many nuclear receptors that act as transcription factors, RARα has two nuclear localization signals (NLS) that are critical for its nuclear targeting [38]. Although mostly confined within the nucleus of neurons during development, the RARα distribution shifts as neurons mature, displaying a gradual increase in cytoplasmic localization [38,58]. A nuclear export signal (NES) in RARα is required for its translocation into the cytosol as RARα bearing a NES mutation is strictly localized to the nucleus [38]. Interestingly, the two NLS are located in domains C and D of RARα, a region where FMRP binds. FMRP contains both NLS and NES signals and shuttles between the nucleus and cytoplasm [59,60]. It is conceivable that FMRP binding masks the NLS in RARα, facilitating its NES-mediated nuclear export. 

Our previous study showed that FMRP is enriched in dendritic RNA granules in cultured hippocampal neurons [42], a prime location for dendritic protein synthesis. Importantly, although RARα is present only at low levels in RNA granules at basal condition, the treatment with RA or synaptic activity blocking reagents (e.g., TTX + APV) that induce de novo RA synthesis within neurons, triggers rapid translocation of RARα into FMRP-rich RNA granules [42]. Following RARα translocation, the synthesis of GluA1 protein ensues in these FMRP-RNA granules [42]. Interestingly, the results in this study show that the RA treatment enhances FMRP-RARα binding, suggesting a scenario that synaptic inactivity-induced RA synthesis facilitates translocation of RARα and its associated target mRNAs (e.g., GluA1 mRNA) into dendritic RNA granules by enhancing FMRP-RARα binding, permitting translational activation of specific mRNA species. This notion is consistent with the observation that although both FMRP-∆RGG- and FMRP-I304N mutants have normal binding with RARα, they do not rescue homeostatic synaptic plasticity in *Fmr1* KO neurons since both mutants are defective in their incorporation into polyribosomes or RNA granules [13,53,54,61,62,63,64].

The connection between FMRP and RARα not only expands the functional repertoire of FMRP to the regulation of homeostatic synaptic plasticity, but also brings up additional synaptic mechanisms that should be taken into consideration when investigating neural mechanisms underlying cognitive defects in FXS. Neurological diseases associated with intellectual disability are often attributed to changes in Hebbian synaptic plasticity because LTP/LTD are considered cellular mechanisms of learning and memory. However, increasing evidence show defective homeostatic synaptic plasticity in various ASD models [29,30,65,66,67,68]. In addition to the disrupted excitation/inhibition (E/I) balance in various ASD models (which may be a consequence of defective homeostatic plasticity) [69,70], defective homeostatic synaptic plasticity has been directly examined and reported in *Mecp2* KO (a Rett syndrome model) [66,67,71], *Fmr1* KO mice [29,30], and FXS human neurons [31]. Might homeostatic synaptic plasticity contribute to learning and memory through interaction with Hebbian plasticity? A recent study using RARα conditional KO mice showed that homeostatic synaptic plasticity indeed significantly impacts Hebbian plasticity and behavioral learning [72]. In human subjects, seizure (resulting from E/I imbalance) and intellectual disability are common symptoms affecting a fraction of patients with ASDs, Schizophrenia, and neurodegeneration disorders. Indeed, some of the disease-relevant biological pathways are interconnected between FXS, ASDs, and Schizophrenia [73,74,75]. Thus, understanding the molecular details of impaired homeostatic synaptic plasticity in ASD models may provide insights into discovering potential targets for therapeutic treatment of ASDs, including ASDs in FXS, and other neurodevelopmental disorders [68]. Those studies should incorporate further refined methodologies [76] to assess whether parameters that may underlay variability in FXS, such as FMRP levels and comorbid diagnosis of ASD [7], have an impact on the response to those potential targeted therapeutics.

## Figures and Tables

**Figure 1 ijms-22-06579-f001:**
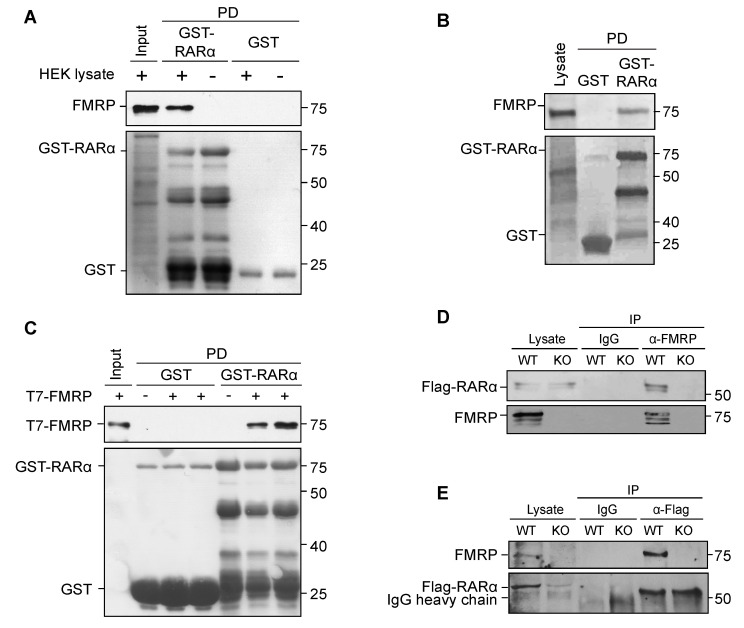
RARα interacts directly with FMRP. (**A**) Pull-down of endogenous FMRP from HEK cell lysate by purified GST-RARα coupled to Glutathione sepharose beads (top panel: FMRP immunoblot; bottom panel: Ponceau staining). Purified GST coupled to Glutathione sepharose beads was used as a negative control. (**B**) Pull-down of endogenous FMRP from P30 mouse hippocampal lysate by purified GST-RARα coupled to Glutathione sepharose beads (top panel: FMRP immunoblot; bottom panel: Commassie staining). Purified GST coupled to Glutathione sepharose beads was used as a negative control. (**C**) In vitro pull-down of purified T7-FMRP by purified GST-RARα but not GST (top panel: T7 immunoblot; lower panel: Ponceau stain). (**D**,**E**) Co-immunoprecipitation of endogenous FMRP with Flag-RARα from DIV14-15 primary hippocampal neurons using anti-FMRP antibodies (**D**) and anti-Flag antibodies (**E**). In both experiments, lentivirus expressing Flag-RARα-IRES-GFP-Cre in RARα^fl/fl^ background was used to replace endogenous RARα with Flag-tagged RARα. Hippocampal cultures obtained from RARα_f_^fl/fl^/*Fmr1* KO mice (KO) were used as negative controls for specificity of pull-down.

**Figure 2 ijms-22-06579-f002:**
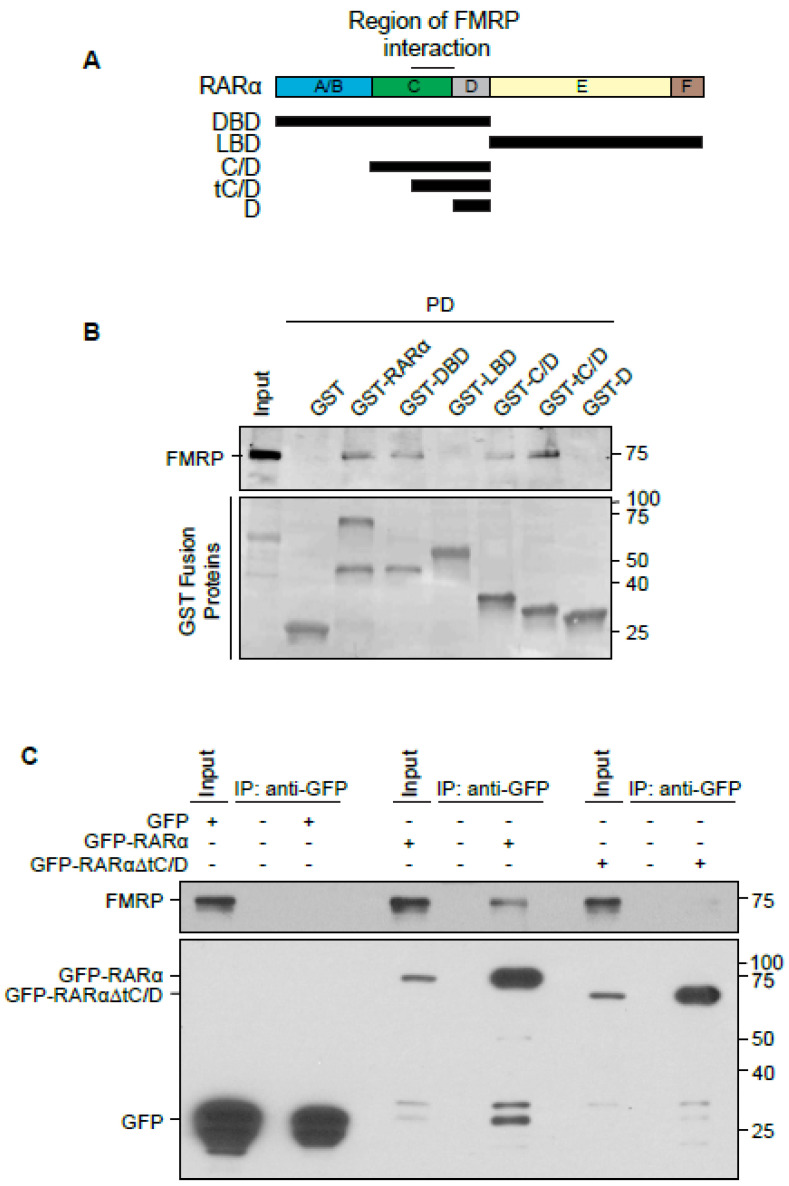
Determination of RARα domains important for the interaction with FMRP. (**A**) Schematics of RARα domain organizations and truncation mutants used for interaction assays. All fusion proteins contain an N-terminal GST tag. (**B**) GST-tagged purified RARα truncation mutants from (**A**) were incubated with HEK cell lysates; pull-down of endogenous FMRP in P30 mouse hippocampal lysates was detected by immunoblot (top panel). Coomassie blue staining shows expression levels of truncation mutants (bottom panel). (**C**) Immunoprecipitation of endogenous FMRP (top panel) in HEK cells transfected with GFP, GFP-RARα, and GFP-RARαΔtC/D (bottom panel) using anti-GFP antibodies.

**Figure 3 ijms-22-06579-f003:**
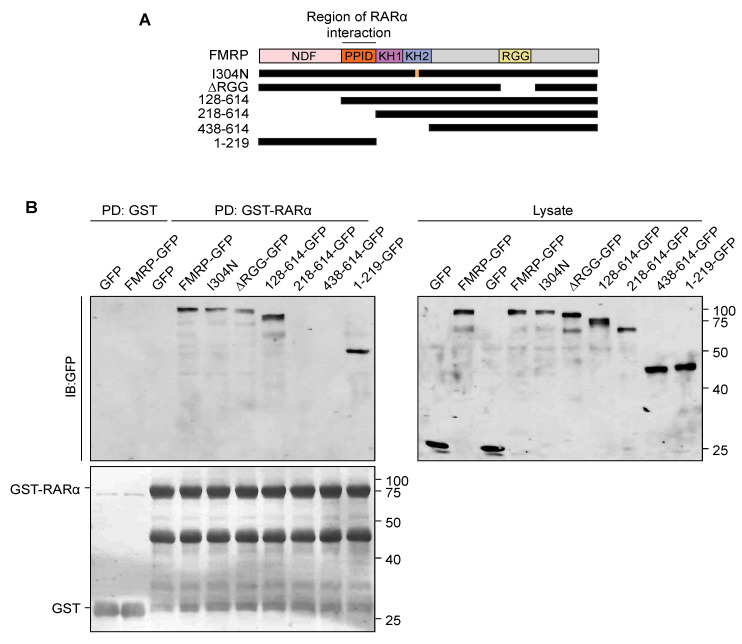
FMRP domains critical for the interaction with RARα. (**A**) Schematics of FMRP domain organization and truncation/point mutations used to determine the RARα binding site. All proteins have a C-terminal GFP tag. Numbering represents the position of amino acid in FMRP. (**B**) Pull-down of FMRP mutants from (**A**) expressed in *Fmr1* KO MEFs with GST or GST-RARα.

**Figure 4 ijms-22-06579-f004:**
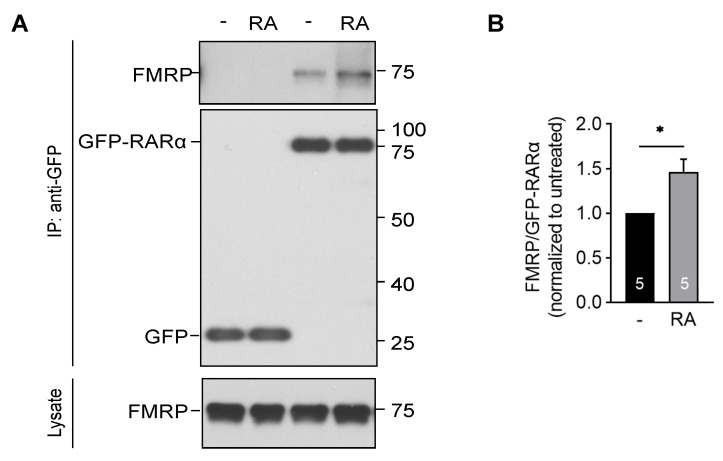
Retinoic acid enhances the interaction between FMRP and RARα. (**A**) Immunoprecipitation with anti-GFP antibodies of FMRP with GFP-RARα in RA treated (1 μM for 1 h) HEK cells. (**B**) Quantification of the amount of FMRP immunoprecipitated by GFP-RARα after treatment with RA (Student’s *t*-test, *n* = 3; * *p* < 0.05). The graph represents average values ± SEM.

**Figure 5 ijms-22-06579-f005:**
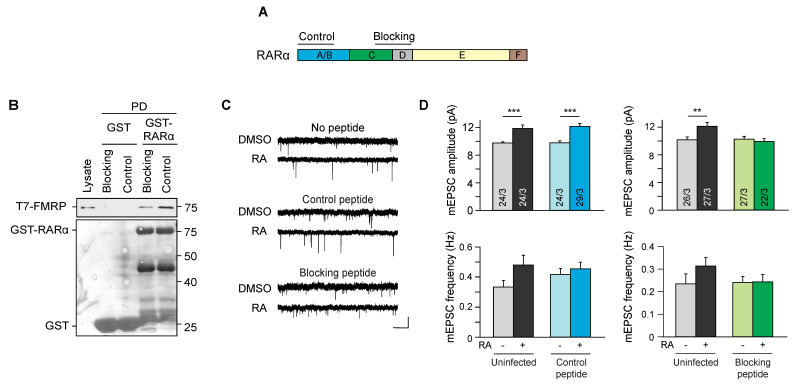
Blocking interaction between FMRP and RARα abolishes RA-induced synaptic plasticity. (**A**) A schematic drawing showing the locations of the blocking peptide and the control peptide mapped to RARα. (**B**) Pull-down of T7-FMRP with GST-RARα in the presence of purified control or blocking peptide. (**C**) Representative traces of miniature EPSCs recorded from CA1 pyramidal neurons of hippocampal slice cultures infected with lentivirus expressing either the GFP-tagged control or blocking peptide and treated with RA. Scale bars, 10 pA, 1 s. (**D**) Quantification of miniature EPSC amplitude and frequency after a 4 h treatment with DMSO or 10 μM RA (** *p* < 0.01; *** *p* < 0.001; Mann-Whitney). All graphs represent average values ± SEM.

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
