# Peer review of "FMRP Interacts with RARα in Synaptic Retinoic Acid Signaling and Homeostatic Synaptic Plasticity"

_ijms, 2021, doi:10.3390/ijms22126579_

Round 1

Reviewer 1 Report

Among the various types of abnormal synaptic function and synaptic plasticity phenotypes reported in FXS animal models, defective synaptic retinoic acid (RA) signaling and subsequent defective homeostatic plasticity have emerged as a major synaptic dysfunction. However, the mechanism underlying the defective synaptic RA signaling in the absence of FMRP is unknown. The authors show that RARa, the RA receptor critically involved in synaptic RA signaling, directly interacts with FMRP. This interaction is enhanced in the presence of RA. Blocking the interaction between FMRP and RARa with a small peptide corresponding to the critical binding site in RARa abolishes RA-induced increases in excitatory synaptic transmission, recapitulating the phenotype seen in the Fmr1 knockout mouse. Taken together, these data suggest that not only are functional FMRP and RARa necessary for RA- dependent homeostatic synaptic plasticity, but that the interaction between these two proteins is essential for proper transcription-independent RA signaling. The results may provide further mechanistic understanding into FXS synaptic pathophysiology.

This is a timely and well-written manuscript that expands on multiple interactions of FMRP. The identified functional FMRP and RARa link necessary for RA- dependent homeostatic synaptic plasticity, and that the interaction between these two proteins is essential for proper transcription-independent RA signaling may advance the field toward another potential targeted treatment. While the study involves a preclinical study of a FXS model, it is worthwhile to mention relevant key clinical implications in FXS.

  1. Line 41, page 1. “Indeed, many of the mRNA targets and interacting proteins of FMRP are known to regulate synapse structure and function [9, 11, 15-19]”.

Suggest expanding it, for example: A key clinical implication is that the FMRP expression in the brain is the ultimate factor determining the severity of the neurobehavioral phenotype in humans with FXS, including autism spectrum disorder (Neurobiology of autism and intellectual disability: Fragile X syndrome, Budimirovic and Subramanian, 2016  DOI: 10.1093/med/9780199937837.003.005; Budimirovic et al., 2020 in Brain Sciences, https://doi.org/10.3390/brainsci10100694)

  1. Lines 472-73, page 14. However, increasing evidence show defective homeostatic synaptic plasticity in various autism spectrum disorder (ASD) models. Reference is needed: consider https://doi.org/10.1159/000330213
  2. Lines 481-83, page 14. Thus, understanding the molecular details of impaired homeostatic synaptic plasticity in ASD models may provide insights into discovering potential targets for therapeutic treatment of ASDs”.

Suggest clarifying/adding on at the end of the above sentence the following “…potential targets for therapeutic treatment of ASDs, including ASD in FXS (Budimirovic et al., 2020; https://doi.org/10.1159/000330213).

  1. Finally, suggest expanding a bit more relevance of clinically-clinical trials study FMRP, ASD. For example:

Those studies should incorporate further refined methodologies (doi: 10.1515/tnsci-2017-0002) to assess whether parameters that may underlay variability in FXS, such as FMRP levels and comorbid diagnosis of ASD (Budimirovic et al., 2020; Budimirovic and Subramanian, 2016), have an impact on response to those potential targeted therapeutics.

Author Response

We thank the reviewers for their positive evaluation of our work. We have made revisions of the text based on their suggestion. Please see detailed responses below.

Reviewer 1

  1. Line 41, page 1. “Indeed, many of the mRNA targets and interacting proteins of FMRP are known to regulate synapse structure and function [9, 11, 15-19]”.

Suggest expanding it, for example: A key clinical implication is that the FMRP expression in the brain is the ultimate factor determining the severity of the neurobehavioral phenotype in humans with FXS, including autism spectrum disorder (Neurobiology of autism and intellectual disability: Fragile X syndrome, Budimirovic and Subramanian, 2016  DOI: 10.1093/med/9780199937837.003.005; Budimirovic et al., 2020 in Brain Sciences, https://doi.org/10.3390/brainsci10100694)

We thank the reviewer for this very informative suggestion. We have added the suggested content in the introduction.

  1. Lines 472-73, page 14. However, increasing evidence show defective homeostatic synaptic plasticity in various autism spectrum disorder (ASD) models. Reference is needed: consider https://doi.org/10.1159/000330213

References added.

  1. Lines 481-83, page 14. Thus, understanding the molecular details of impaired homeostatic synaptic plasticity in ASD models may provide insights into discovering potential targets for therapeutic treatment of ASDs”.

Suggest clarifying/adding on at the end of the above sentence the following “…potential targets for therapeutic treatment of ASDs, including ASD in FXS (Budimirovic et al., 2020; https://doi.org/10.1159/000330213).

We have modified the text accordingly. 

  1. Finally, suggest expanding a bit more relevance of clinically-clinical trials study FMRP, ASD. For example:

Those studies should incorporate further refined methodologies (doi: 10.1515/tnsci-2017-0002) to assess whether parameters that may underlay variability in FXS, such as FMRP levels and comorbid diagnosis of ASD (Budimirovic et al., 2020; Budimirovic and Subramanian, 2016), have an impact on response to those potential targeted therapeutics.

We have modified the text and added additional references.

Reviewer 2 Report

In the present study, the Authors investigated the physical interaction between fragile X mental retardation protein (FMRP) and retinoic acid (RA) receptors (RARα) and found that RARα directly binds to FMRP, and that the binding is enhanced in the presence of RA. They also demonstrated that this direct interaction might be critical for RA-induced homeostatic synaptic plasticity.

Overall, I found the present study, timely, well conducted, very interesting and scientifically sound: enjoyed reading it! I have only some minor comments aimed to improve the high quality of the paper and these are outlined below:

1) The Fragile X syndrome is known to be frequently associated with several psychiatric disorder such as schizophrenia (especially early-onset first episode) and other psychoses. I believe that this point deserves a brief note in the Introduction with appropriate references (see dois: 10.1093/schbul/sbt153 and 10.3389/fgene.2019.00268).

2) As the Authors established that FMRP-RARα interaction is required for RA-induced synaptic plasticity, I guess if BDNF or NGF might be involved in some ways. Please, discuss this point or add Your opinion about it (see Martinotti et al. J Biol Regul Homeost Agents. 2012 Jul-Sep;26(3):347-56).

3) In a translational way, what it can be drawn from this study in terms of human implications and future research directions?

Author Response

We thank the reviewers for their positive evaluation of our work. We have made revisions of the text based on their suggestion. Please see detailed responses below.

Reviewer 2

  • The Fragile X syndrome is known to be frequently associated with several psychiatric disorder such as schizophrenia (especially early-onset first episode) and other psychoses. I believe that this point deserves a brief note in the Introduction with appropriate references (see dois: 1093/schbul/sbt153 and 10.3389/fgene.2019.00268).

We thank the reviewer for pointing this out. We think this is more appropriate for the discussion part of the paper and have added relevant text and references in the last paragraph of discussion.

  • As the Authorsestablished that FMRP-RARα interaction is required for RA-induced synaptic plasticity, I guess if BDNF or NGF might be involved in some ways. Please, discuss this point or add Your opinion about it (see Martinotti et al. J Biol Regul Homeost Agents. 2012 Jul-Sep;26(3):347-56).

We added one sentence and references in discussion pointing out the converging biological pathways in FXS, ASDs and Schizophrenia.

  • In a translational way, what it can be drawn from this study in terms of human implications and future research directions?

See additional discussion added in the last paragraph.